# Plant Essential Oils as a Tool in the Control of Bovine Mastitis: An Update

**DOI:** 10.3390/molecules28083425

**Published:** 2023-04-13

**Authors:** Alice Caneschi, Anisa Bardhi, Andrea Barbarossa, Anna Zaghini

**Affiliations:** Department of Veterinary Medical Sciences, Alma Mater Studiorum—University of Bologna, Via Tolara di Sopra 50, 40064 Bologna, Italy; anisa.bardhi@unibo.it (A.B.); andrea.barbarossa@unibo.it (A.B.); anna.zaghini@unibo.it (A.Z.)

**Keywords:** essential oils, bovine mastitis, antibacterial activity, antimicrobial resistance (AMR)

## Abstract

Bovine mastitis is a major concern for the dairy cattle community worldwide. Mastitis, subclinical or clinical, can be caused by contagious or environmental pathogens. Costs related to mastitis include direct and indirect losses, leading to global annual losses of USD 35 billion. The primary treatment of mastitis is represented by antibiotics, even if that results in the presence of residues in milk. The overuse and misuse of antibiotics in livestock is contributing to the development of antimicrobial resistance (AMR), resulting in a limited resolution of mastitis treatments, as well as a serious threat for public health. Novel alternatives, like the use of plant essential oils (EOs), are needed to replace antibiotic therapy when facing multidrug-resistant bacteria. This review aims to provide an updated overview of the in vitro and in vivo studies available on EOs and their main components as an antibacterial treatment against a variety of mastitis causing pathogens. There are many in vitro studies, but only several in vivo. Given the promising results of treatments with EOs, further clinical trials are needed.

## 1. Introduction

### 1.1. Bovine Mastitis

Bovine mastitis is the inflammation of mammary gland parenchyma due to physical trauma or microorganism infections which results in the altered physical, chemical, and bacteriological quality of milk [1]. Due to the reduced yield and poor quality of milk, bovine mastitis is the most common disease that leads to economic loss and is a major concern for the dairy cattle community worldwide [2,3,4]. Bovine mastitis depends on a combination of animal-, environmental-, and pathogen-related factors; furthermore, it can be classified into three categories based on the degree of inflammation, namely clinical, sub-clinical, and chronic mastitis [5,6]. Clinical bovine mastitis is evident and easily detected by visible abnormalities in the milk, swelling or tenderness of the udder, and general symptoms such as fever and loss of appetite in dairy cows [7]. On the other hand, no overt signs are observed in subclinical mastitis, but there is a decrease in milk production and a change in its composition [8]. Chronic mastitis is an inflammation that lasts for several months, with clinical relapses.

The invasion of the teat canal by pathogenic organisms, including bacteria, fungi, and viruses, is considered to be the main cause of bovine mastitis [9]. The most common mastitis-causing organisms are bacteria and based on the bacterial origin the infection can be classified into two types: contagious and environmental [10]. Contagious pathogens such as *Staphylococcus aureus*, *Mycoplasma* spp., and *Streptococcus agalactiae*, usually live on the cow’s udder and teat skin, colonizing and growing into the teat canal [11]. These microorganisms are capable of establishing sub-clinical infections, usually with an elevation in the somatic cell count (SCC). Contrarily, environmental pathogens are not part of the mammary gland microbiota—they are opportunistic invaders—and are not adapted to survive within the host like contagious pathogens. After invading the mammary gland, environmental pathogens multiply, stimulate the host response system and are rapidly eliminated [12]. The environmental pathogens include mainly *Streptococcus* spp. (e.g., *S. uberis*), coliforms species (e.g., *E. coli*, *Klebsiella* spp., *Enterobacter* spp.), *Pseudomonas* spp., etc. [13]. These pathogens cause a less manageable mastitis, because infection of the mammary gland occurs primarily apart from the milking procedure.

Bacteria causing mastitis, such as *S. uberis* and *S. aureus*, are also able to produce biofilm [14]. The biofilm matrix is constituted of polysaccharides, microbial cells, water and additional extra-cellular products, which supply a protective and repaired environment that allows for bacterial growth [15]. As a result, bacteria are more resistant to antibiotics and host defences. Hence, the ability of pathogens to produce biofilms can lead to difficulties in treating recurrent infections [16]. Biofilm also enables bacteria to adapt to unfavorable environment [17].

Correct diagnosis of mastitis is essential in the dairy industry, both for public health and economic reasons, and for animal welfare. To prevent the onset of the disease, the diagnosis must be early, accurate, and rapid, as must management and treatment. There are two types of diagnostic tests, conventional and advanced. Conventional diagnostic tests are usually qualitative, with lower specificity and sensitivity, while advanced tests are quantitative, highly specific, and sensitive [18].

Diagnosis involves two steps, the determination of the presence or absence of the disease and then the identification of the aetiological agent [19]. In the case of clinical mastitis, the diagnosis is less complex because disease status is indicated by the swollen quarters/udder and poor milk quality; both can be detected by farmers [20]. Subclinical mastitis cannot be visually diagnosed and on-farm screening tests are used traditionally, such as somatic cell count (SCC), California mastitis test, and Surf field mastitis test [21,22].

Given the wide range of pathogens that cause mastitis, early and accurate identification of the bacteria involved is crucial for both control of the disease’s spread (contagious organism) and proper antibiotic selection for therapeutic purposes [7].

Nowadays, thanks to the improvement in sensitivity and specificity, PCR is considered the gold standard, while in the past microbial culture has been used [23].

### 1.2. Economic Implications of Bovine Mastitis Worldwide

Bovine mastitis represents a major disease in the dairy industry and causes global annual losses of approximately USD 35 billion [4]. Because the disease usually induces permanent and irreversible damage to milk-producing glandular tissues, the costs related to mastitis include direct and indirect losses like those from discarded milk, drugs, increased labor and therapy costs, and premature culling of cows [24,25,26].

An estimation has been shown for the total cost per mastitis cow per year in Europe dairy farms to be an average of EUR 124 [27,28], with an incidence varying from 13 to 40 cases/100 cow-years [29], although the production losses and expenditure associated with mastitis are generally underestimated and potentially miscalculated [2].

Subclinical mastitis is steadily increasing and countries in which disease prevalence is high, such as Ethiopia, Kenya, South Africa, and Uganda, display a high prevalence of subclinical mastitis of between 60 and 80% [2]. Moreover, losses due to subclinical mastitis are three times more severe than those due to clinical cases [30], because they are difficult to quantify as they are not visible to the farmers [31]. Subclinical mastitis has a negative outcome on the quality of milk, resulting in diminished milk proteins, fat, lactose, and electrolytes [32].

### 1.3. Treatment and Alternatives

The National Institute for Research in Dairying (NIRD) introduced a five-point plan that is effective in controlling contagious mastitis pathogens. The five points are: (1) identification and treatment of clinical cases; (2) disinfection of teats after milking; (3) dry cow therapy (DCT); (4) culling of chronic cases; and (5) regular cleaning of the milking machine [33]. However, this plan is not successful against environmental pathogens, and therefore, should be combined with other appropriate strategies to prevent mastitis infections [34].

The primary treatment of mastitis is the use of antibiotics, such as penicillin, ampicillin, tetracycline, gentamicin, etc., which can be administered by intramammary, intramuscular, or intravenous infusions [35]. Dry cow therapy is one of the most effective options for controlling and inhibiting mastitis progression. Immediately after the last milking, an intramammary shot of antibiotic is injected into the cow’s udder via the teat canal, then the teat is sealed with a plug, which acts as a physical barrier against bacterial invasion. Ideally, treatment should last enough to cure subclinical mastitis and be brief enough to not lead to antibiotic resistance. As there is no milk production, the dry period is the best time to treat mastitis, since the risk of incorporating the antibiotic into the food chain is minimized [36].

When an active mastitis infection is detected, the first thing to do is to milk out the cow to completely remove any milk clots, debris, bacteria, and toxins. The antibiotic is then infused intramammarily—so that it reaches the udder—or systematically [35]. In some acute cases, parenteral administration together with intramammary infusion is suitable on the advice of the veterinarian, resulting in a higher cure rate [37,38]. Long-acting antibiotics are not suitable for mastitis detected during lactation, as the main objective is to bring the cow back to milking.

Although the use of antibiotics is still the primary approach for treatment, its efficacy is limited and bacteriological elimination is often less than 60% [39]; additionally, the development of antibiotic-resistant strains of pathogens has become a critical challenge in antibiotic treatment [40].

In the Regulation (EU) 2019/6 of the European Parliament and of the Council of 11 December 2018 on veterinary medicinal products and repealing Directive 2001/82/EC, it is established that “Antimicrobial medicinal products shall not be used for prophylaxis other than in exceptional cases, for the administration to an individual animal or a restricted number of animals when the risk of an infection or of an infectious disease is very high and the consequences are likely to be severe” in order to prevent the spread of antimicrobial resistance.

Antimicrobial resistance (AMR) is defined as the ability of microorganisms to survive or grow in the presence of a concentration of an antimicrobial (meaning antibiotics) agent that is generally sufficient to inhibit or kill microorganisms of the same species [41].

The consistent use of antibiotics, sometimes misused or overused, builds a selective pressure on both pathogenic bacteria and commensal bacteria, such as gut or skin microbiota, that can result in the prevalence of antibiotic-resistant bacteria, and the spread of resistant bacteria or genes. Nowadays, AMR is a widespread and growing phenomenon and it has a serious impact on society, both economically and in terms of healthcare [42]. It represents a global crisis and one of the most complex challenges, already responsible for 700,000 deaths a year globally that could rise to 10 million by 2050 [43,44], resulting in 11% loss in the production of livestock [45].

In animals, AMR is much more complex than it is in humans and requires an even more attentive and conscious use of antibiotics [42]. In particular, the use of antibiotics, like any other drug, in food animals can result in the presence of residues in edible tissues, as well as in milk [46]. Most veterinarians are aware of AMR, but veterinary antimicrobial stewardship is a relatively new concept in veterinary medicine that needs to be further consolidated and pursued.

Hence, it is necessary to look for alternative treatments based on natural products such as plants, especially when facing multidrug-resistant bacteria [47]. Therefore, researchers all over the world are working to find novel alternatives to the use of antibiotics, such as nanotechnology [48], bacteriophage therapy [49], antimicrobial peptides (AMPs) [50], probiotics [51], immunotherapy [52], and medicinal plants [53] among others.

The use of plant extracts is documented in traditional medicine around the world and there is a growing interest in their application for the treatment of bovine mastitis. Many medicinal plants with different biological properties are already used in this field, although in vivo trials reporting their efficacy, pharmacokinetics and pharmacodynamics are scarce [54]. Due to the presence of a large variety of constituents, it is more difficult for microorganisms to acquire resistance against plants extracts or essential oils. Moreover, plant extracts can exhibit antibacterial activity against both resistance acquired and non-resistance pathogens [55]. Therefore, understanding the specific mechanism behind phytochemical action with rigorous experimental studies is vital to develop new antimicrobial agents [56].

## 2. Essential Oils

The International Organization for Standardization (ISO) defines essential oils (EOs) as “products obtained from a natural raw material of plant origin, by steam distillation, by mechanical processes from the epicarp of citrus fruits or by dry distillation, after separation of the aqueous phase–if any–by physical processes” [57]. Essential oils are generally volatile liquids, but sometimes solids, extracted from various parts of the aromatic plants, including the aerial parts, usually made up of flowers, leaves, and stems (chamomile, peppermint, lavender); bark (cinnamon); fruits (anise); seeds (nutmeg); as well as in the radix and rhizomes (curcuma and ginger) [58]. Essential oils are secondary metabolites produced by aromatic plants, and they play an important role in the protection of plants as antibacterials, antivirals, antifungals, insecticides, and additionally, against herbivores by reducing their palatability for such plants. Furthermore, EOs may attract some insects to favor the dispersion of pollens and seeds, or repel undesirable others [59]. At present, almost 3000 essential oils have been described and 300 of these are commercially important especially for the pharmaceutical, agronomic, food, sanitary, cosmetic, and perfume industries [58]. The medicinal uses of essential oils, as well as of herbal remedies produced from plants containing essential oils, have long been documented (more in human than in veterinary medicine) for conditions that can benefit from their antimicrobial, anti-inflammatory, bronchodilatory, expectorant, anticonvulsant, cholagogic, analgesic, and spasmolytic activity [60].

Essential oils can vary in quality, quantity and in composition according to climate, soil composition, plant organ, plant age, and vegetative cycle stage, and, finally, type of extraction. To obtain a constant composition, EOs should be extracted under the same conditions, from the same organ of the plant which has been growing on the same soil, under the same climate and has been picked in the same season. For this reason, most of the commercialized essential oils are chemotyped by mass spectrometry and gas chromatography analysis [59].

Essential oils are very complex natural mixtures that can contain 20–100 different plant secondary metabolites belonging to a variety of different chemical classes, like aromatic alcohols, acids, esters, ketones, phenolics, aldehydes, and hydrocarbons [61,62]. Essential oils are characterized by two or three major components, mainly terpenes, terpenoids, and phenylpropanoids, at rather higher (20–70%) concentrations than other components present in trace amounts [59]. In some cases, the main components are hydrocarbons, whereas in others, the major derivatives are oxygenated. In a small number of plant species, the predominant derivatives are aromatic principles (e.g., thyme with thymol and carvacrol, and peppermint with menthol) [58]. Usually, the major components determine the biological properties of the EO.

Thanks to their promising biological effects, in recent years, there has been a growing interest in researching and exploring EOs, and/or their derived components, as candidates to fight against bacterial infections and reduce AMR [63,64,65].

### Antibacterial Effect

The activity of EOs on bacteria can be both bacteriostatic (blocking the growth of bacteria) or bactericidal (killing the bacterial cells). The therapeutic efficacy of essential oils may have a different outcome mainly depending on the different target bacteria (Gram-negative or Gram-positive) and on bacterial infections (acute, chronic, more or less severe) [66].

The lipophilic nature of EOs and their components allows them to cross the cell membrane of bacteria and mitochondria, destabilizing cellular architecture, leading to disruption of membrane integrity and increased permeability, which causes the loss of many cellular activities, among them the processes of membrane transport, energy production, and other metabolic regulatory functions [65].

The antibacterial effects of EOs’ are linked to leakage of cellular components and loss of ions, due to the increased membrane permeability, as well as to reduce membrane potential, the disruption of proton pumps, and the depletion of ATP, leading to cell lysis and cell death [67,68,69]. Essential oils were found to be active against bacteria overexpressing efflux pumps and thus developed tolerance toward antibiotics [70].

By decreasing membrane potential, essential oils cause depolarization of mitochondrial membranes, and moreover, they affect Ca++ ionic cycling as well as other ionic channels. Lastly, EOs alter the proton pump and ATP pool and reduce the pH gradient. Permeabilization of outer and inner mitochondrial membranes leads to cell death by apoptosis and necrosis [71]. Starting from the cell wall or outer cell membrane, chain reactions invade the entire cell, through the membranes of various organelles such as mitochondria and peroxisomes. A phenol-type prooxidant activity is suggested by these effects [59].

While sometimes the overall effect of an EO cannot be attributed to any of the major constituents, since the combination of molecules modifies their activity to exert a significant impact [61], in most cases, the bioactivity of a particular EO is determined by one or two of its major components [59]. Active constituents are mainly represented by isoprenes, such as monoterpenes, sesquiterpenes, and related alcohols, along with other hydrocarbons and phenols. In particular, terpene hydrocarbons and oxygenated terpenes show marked antibacterial activity [72].

Phenolic compounds such as eugenol, carvacrol, and thymol from various plant origins are highly active against many microorganisms. The hydrophobic benzene ring and aliphatic side chains of thymol sink into the inner part of the biological membrane, while the hydrophilic part of the molecule interacts with the polar part of the membrane. This is responsible for huge changes in the membrane structure, such as destabilization of the lipid layer, decrease in elasticity, and increase in fluidity [73]. The importance of the hydroxyl group in the phenolic structure has been confirmed by comparing the activity of carvacrol with its methyl ether form. Moreover, the relative position of the hydroxyl group influences the effectiveness of the terpenes as the two isomers, thymol, and carvacrol, showed different activities against Gram-positive and Gram-negative bacteria [69].

Since the outer cell membrane of Gram-negative bacteria possesses hydrophilic properties, the contact between the hydrophobic constituents of EOs and the bacterial cell of this bacteria is more difficult. In contrast to this, EOs are able to damage directly the cell membrane of Gram-positive bacteria, resulting in the rupture of cell membrane by blockade of the enzymes system and progressivity of ion permeability [65].

The high incidence of bovine mastitis and its related costs represent a major concern for the dairy cattle community worldwide. Although the use of antibiotics is still the primary approach for treatment, its efficacy is limited and bacteriological elimination is not granted. Since the overuse and misuse of antibiotics in livestock are contributing to the development of AMR, there is a need for sustainable antimicrobials that can fight resistant microorganisms and inhibit the dissemination of AMR. Due to their antibacterial effect, EOs and/or their derived components have gained much attention in this field.

Given these promising results, this review will focus on the role of essential oils when used in the control of bovine mastitis.

## 3. EOs and Their Components with Reported In Vitro Antibacterial Activity

The aim of the present report is to collect the currently available data on the main effects of EOs against mastitis causing pathogens. We conducted a search for the literature primarily describing the antibacterial effects, either in vitro (see Table 1) or in vivo, of EOs and their principal components. The search terms “(essential oil AND bovine mastitis) OR (essential oil AND mastitis)” were used to query PubMed, CAB Direct, and Google Scholar (for details, see Appendix A).

Relevant publications were selected and analyzed. Studies conducted on the most commonly used plants EOs have been summarized in Table 1.

### 3.1. Cinnamon

Cinnamon, a plant native to Sri Lanka, is an important popular spice of tropical Asia, obtained from the inner bark of various trees, found worldwide, belonging to the genus *Cinnamomum* (about 250 species). The most significant volatile oils in cinnamon come from *C. burmannii*, *C. camphora*, *C. cassia*, *C. osmophloeum*, *C. verum*, and *C. zeylanicum* [101]. Antibacterial properties associated with cinnamon are due to its characteristic secondary metabolites such as trans-cinnamaldehyde, cinnamyl acetate, eugenol, L-borneol, camphor, caryophyllene oxide, β-caryophyllene, and L-bornyl acetate [102] (Figure 1).

Several in vitro studies showed that different cinnamon EOs have bactericidal effects against bovine mastitis pathogenic isolates [77,78,79,80,81,82,83], as well as antifungal [80,83] and anti-algae [84] activity.

Essential oils from *Cinnamomum aromaticum* [80], *Cinnamomum cassia* [82,83], and *Cinnamomum zeylanicum* [77,78] showed good antibacterial activity against *Staphylococcus* spp. isolated from mastitis milk. The higher MIC and MBC values in *E. coli* isolates than in *Staphylococcus* spp. indicate that cinnamon oil is more effective against Gram + than Gram-. The difference in efficacy is likely due to the outer membrane of Gram-, which acts as an impermeable barrier, thus preventing cinnamon oil from penetrating into cells [81].

Cinnamon oil showed an in vitro bactericidal effect against *S. agalactiae* and a synergistic interaction when combined with sylver nanoparticles, resulting in 100% inhibition of bacterial cells within 4 h [79]. Furthermore, cinnamon oil was able to exhibit a potent inhibitory activity against biofilm formation of *S. agalactiae*, causing variable degrees of damage to the surface of cultures and a remarkable down-regulation of pili biosynthesis genes (*pilA* and *pilB*), as well as their regulator (*rogB*) [79].

*C. cassia* oil led to the loss of membrane impermeability in both *S. aureus* and *E. coli* by causing deformities with ruptured membrane, which was associated with decreased ATP synthesis, possibly due to disruption of membrane potential cells [81].

In a time–kill assay with cinnamon oil, the number of bacterial cells decreased during the incubation time and the complete elimination of live cells from both suspensions containing cinnamon oil at MIC and 2 × MIC was achieved after 8 h of incubation [82].

A sub inhibitory concentration of *C. cassia* oil exert also an anti-quorum sensing effect, repressing the production of AI-2, a universal signal molecule that mediates quorum sensing, in *S. aureus* and *E. coli* isolates, providing an additional layer of protection to bovine mastitis [81].

The high antibacterial and antibiofilm activity of cinnamon oil certainly depends on its high cinnamaldehyde content as demonstrated by Budri et al. [77], where cinnamaldehyde had lower MIC than *Cinnamomum zeylanicum*, and it significantly reduced biofilm formation.

Additionally, trans-cinnamaldehyde, an unsaturated aldehyde, was shown to be responsible for antibacterial activity [74,78]. In these molecules the acrolein group is essential for the activity [101].

Furthermore, cinnamon may reduce both inflammation and damage of the mammary tissue associated with bovine mastitis disease [103].

### 3.2. Origanum

A member of the Lamiaceae family, *Origanum* is a genus of important medicinal and aromatic plants, comprising various species with characteristic smell and taste of flowers and leaves [104]. These species are mainly distributed in the Mediterranean, North African, Euro-Siberian, and Iranian-Siberian regions [105].

The major constituents of *origanum* EO are several terpenes, phenols, phenolic acids and flavonoids, with a predominance of carvacrol and thymol (with fair amounts of p-cymene and terpinene) or terpinen-4-ol, linalool, and sabinene hydrate [104] (Figure 2). The EO is responsible for various pharmacological activities, such as antimicrobial activity, and carvacrol is usually the main component [76,83,91,94,95,96]. However, in some *origanum* species, thymol is considered one of the main phytoconstituents responsible for the biological activity. For example, *Origanum floribundum* Munby demonstrates a high thymol content (50.47%) in its essential oil [92].

Carvacrol and thymol are structural isomers with a hydroxyl group positioned at different positions relative to the phenolic ring and represent ones of the most common and well-known terpenoids, namely terpenes with added oxygen molecules or that have had their methyl groups moved or removed by specific enzymes [106].

Carvacrol is a phenolic monoterpenoid, primarily found in the EO of oregano, that acts on microbial cells, causing structural and functional damage to their membranes and resulting in increased permeability. It can affect the outer membrane of Gram-negative bacteria with disintegrating effects, as well as alter the transport of ions [107]. The mode of action of carvacrol appears to be to enhance membrane fluidity and permeability [108].

In a study against mastitis causing multi-drug-resistant pathogens, carvacrol (CAR) showed strong inhibitory activity against *S. aureus* (zone of inhibition: 32.7 ± 3.01 mm) and *E. coli* (31.7 ± 0.58 mm), but showed moderate activity against *K. pneumoniae* (13.7 ± 0.58 mm) [74]. In the same study carvacrol exhibited an additive interaction with some fatty acids, and the more significant combination that took least time to completely kill about 10^6^ CFU/mL bacterial cells was with octanoic acid (OA). Scanning electron microscope revealed that cells exposed to CAR + OA show severe morphological changes and disruption of cell wall and cell membrane, which results in the release of inner cell material into the surrounding environment. The results on the antimicrobial activity of carvacrol are confirmed by Corona-Gómez et al. [75].

Carvacrol is also considered the main responsible for the antioxidant activity of *O. vulgare* [94].

*Origanum vulgare* is the most popular Origanum, popularly called “oregano” in most European countries [109]. *O. vulgare* showed to be effective against predominant mastitis pathogens: *Streptococcus* spp., *E. coli*, *Cronobacter sakazakii*, *Klebsiella oxytoca*, *Staphylococcus aureus*, *Staphylococcus* spp. *coagulase negative*, *Streptococcus dysgalactiae*, and *Streptococcus uberis* [76,94], two environmental causative agents of mastitis, *Serratia marcescens* and *Proteus mirabilis* [95], as well as against methicillin-resistant *Staphylococcus aureus* (MRSA) [91] and *Prototheca zopfii* [93]. Thus, *O. vulgare* can be considered a potential agent for antibacterial treatments against antibiotic-resistant bacterial infections.

*Origanum majorana* (marjoram) is used in traditional and folk medicines for several conditions of the gastrointestinal, respiratory, cardiac, and nervous systems. From marjoram essential oil, chemical compounds such as hydrocarbon monoterpenes, oxygenated monoterpenes, and phenolic compounds have been isolated [86]. *O. majorana* exerts antibacterial activity against three pathogens: *Staphylococcus aureus*, *Streptococcus agalactiae*, and *Escherichia coli*, when tested inside milk as an in vitro model to mimic the udder environment, and it significantly reduced the bacterial population after 4 h. *O. majorana* was reported to inhibit also the growth of *P. zopfii* strains [93].

The EO of *O. floribundum* Munby demonstrated an in vitro appreciable anticandidal activity, with 80% MIC between 17.18 and 23.14 mg/mL, against clinical isolates of *Candida albicans* [92].

### 3.3. Thymus

Another member of the family Lamiaceae, the genus *Thymus*, contains more than 200 species and subspecies worldwide. The Mediterranean area, particularly the Iberian Peninsula and northwestern Africa, is the origin of the genus *Thymus*, from where it has spread to Europe, Greenland, North America, and Abyssinia, as well as Asia. Of these species, *T. vulgaris* L. (common thyme) is the most notable and has long been used. Other species, including *T. zygis* L. (Spanish thyme), *T. serpyllum* L. (wild thyme), and *T. pulegioides* L. (big thyme) are also the major commercial varieties in the world [110].

As aromatic herbs, thyme plants can produce plenty of volatile oils, which are the most abundant secondary metabolites in this genus. Monoterpenes are the prevalent chemical class contained in *Thymus* genus EOs, while sesquiterpenes tend to be less important. The predominant monoterpene components are thymol, carvacrol, p-cymene, α-pinene, linalool, 1,8-cineole, γ-terpinene, camphene, α-terpinene, β-pinene, and terpinen-4-ol [73] (Figure 3).

*Thymus* sp. has been successfully tested against mastitis causing bacteria, both contagious [26,76,82,98,99,100] and environmental [26,95,98,99,100], yeast [92], and algae [84,93,96].

Kovačević et al. [99] demonstrated a stronger antibacterial potential of common thyme EO against *Streptococcus* spp. (β-hemolytic), *Streptococcus* spp., *Enterobacter sakazakii*, *E. coli*, and *Klebsiella oxytoca* when compared to the wild thyme EO. However, wild thyme EO showed a stronger antibacterial activity against *Staphylococcus* spp. (coagulase-negative) and *Staphylococcus* spp. The notable difference between the tested EOs can be attributed to the different content of p-cymene in common thyme (23.83%) and wild thyme (16.66%), while the content of thymol in the wild thyme EO is higher (54.17%) than in the common thyme EO (45.22%).

Thymol belongs to the phenolic monoterpenoid; its structure is similar to that of carvacrol, with hydroxyl groups occupying different positions on the phenolic ring. The antimicrobial activity of thymol, like carvacrol, results in structural and functional changes of the cytoplasmic membrane affecting outer and inner membranes. Thymol can also interfere with intracellular targets and membrane proteins. Its interaction with the membrane affects the permeability and results in the release of ATP and K^+^ ions [106]. Based on its MIC (0.05–0.9 mg/mL), MBC (0.2–1.9 mg/mL), and zone of inhibition (13.5–43.8 mm), thymol is highly effective against isolates from mastitis cows [74,75,76,97]. These studies confirm the importance of thymol within the essential oils for the antibacterial properties of the genus *Thymus*. However, *T. fontanesii* essential oil, rich in carvacrol (62.25%), exerts also a high antibacterial activity, represented by large areas of inhibition, against *E. coli* and *Staphylococcus aureus* responsible for bovine mastitis [98]. The presence of different aromatic, phenolic, and flavonoid compounds are also considered the main reason for the significant antioxidant potential of these plants [99]. Thymol and carvacrol exhibited additive interactions against Gram-negative bacteria and *C. albicans* [75], while the combination with tilmicosin had an addictive effect against *Staphylococcus* spp., *Streptococcus* spp., and *E. coli* [97].

### 3.4. Melaleuca

The genus *Melaleuca* (family Myrtaceae) is native to Oceania, where these plants have been used for centuries by aborigines in traditional Australian medicine, especially for their broad-spectrum antimicrobial activity.

The most important representative agent of this genus is *Melaleuca alternifolia* (Maiden & Betche) Cheel [111]. The essential oil of the plant is known as tea tree oil (TTO) and it shows a long history of medicinal uses. TTO contains more than 100 components, including various monoterpenes, sesquiterpenes, and aromatic compounds [112]. In total, 80–90% of the oil is represented by the monoterpenes terpinen-4-ol, α-terpinene, 1,8-cineole, p-cymene, α-terpineol, α-pinene, terpinolene, limonene, and sabinene (Figure 4). The most abundant (30%) of these is terpinen-4-ol, which plays an important role in the antimicrobial activity of the oil. The quantity of each terpene in TTO can differ greatly depending on the Melaleuca population and chemotype used, climate, age of the leaves, maceration of the leaves, and duration of distillation [111].

Tea tree oil exerts antibacterial action against *E. coli* and *S. aureus*, destabilizing potassium ion gradients, inhibiting respiration, and promoting membrane leakiness, and so far, development of multistep antibiotic resistance in the presence of tea tree oil or terpinen-4-ol has not been observed to have any significant impact [112].

The growth, biofilm formation, and invasion of bovine mammary epithelial cells (BMECs) of *S. aureus* are reduced from addition of TTO, suggesting that TTO can be considered a useful tool for helping antibiotic agents to cure the persistent and chronic infections [113].

Tea tree oil showed bactericidal activity against *Staphylococcus* spp., *Streptococcus* spp., *Escherichia coli*, *Klebsiella pneumoniae*, and *Candida albicans* [75], as well as the algae *Prototheca zopfii*, *P. wickerhamii* [87].

While increasing the secretion of the anti-inflammatory cytokines IL-4 and IL-10, TTO reduces the secretion of IL-2. Therefore, TTO can also provide effective protection to the body by decreasing the proliferation of inflammatory cells without affecting their ability to secrete anti-inflammatory cytokines [112].

The study conducted by Zhan et al. [113] revealed that concentrations of 0.025% and 0.05% TTO increased the proliferation of BMECs and their viability when exposed to *S. aureus*, while 0.1% TTO inhibited the viability of BMECs. Incubated with BMECs, TTO plays a positive role in relieving the inflammatory responses triggered by *S. aureus* and enhances both the IL-8 expression to recruit the polymorphonuclear leukocytes (PMNL) to the mammary gland and the PMNL ability to migrate in *S. aureus*-exposed BMECs.

After TTO supplementation, the proportion of normal BMECs stimulated by lipopolysaccharide (LPS) increased, and the percentage of early apoptosis, late apoptosis, and dead cells dropped. These results suggest that TTO supplementation could help to relieve inflammation at least in part through the reduction of pro-inflammatory cytokines caused by high concentrations of LPS [114].

It should be noted that, despite the remarkable beneficial activities, some toxicity has been observed with TTO when ingested. The LD50 for TTO in a rat model ranges from 1.9 to 2.6 mL/kg bw. TTO can cause both irritant and allergic reactions when applied on the skin at concentrations higher than 10%. Moreover, TTO is stated to be toxic to cats, since 1.5 to 3 teaspoons of TTO can be lethal to 50% of the animals [112].

*Melaleuca armillaris* EO has synergic effect with rifaximin [88] and erythromycin [63] against *S. aureus*. The main component, 1.8-cineole, can disintegrate the cell membrane and reduce the cytoplasm, causing damage to the structure of *S. aureus.* Erythromycin, when using alone, had bacteriostatic activity, but when it was combined with the EO it behaved as a bactericidal antibiotic. The destabilization in the membrane, cell wall, and an eventual decrease in the activity of efflux pumps caused by the EO can facilitate the access of erythromycin into the bacterial cell, where the antibiotic would become more concentrated, facilitating interaction with its site of action.

### 3.5. Lavandula

*Lavandula* is a genus of 39 species of plants, from the family of Lamiaceae, native to the Mediterranean region, growing in natural sites of the lower parts of the mountains [115]. Lavender, especially *L. angustifolia*, represents a medicinal plant, traditionally utilized to overcome diseases, for its antimicrobial and antioxidant activities [116].

Essential oil, for which lavender is mainly known, can be obtained from the flowers by hydrodistillation or steam distillation. More than a hundred components have been identified in the EO, the main ones are: linalool (up to 68.8%) and linalyl acetate (up to 59.4%) (Figure 5). The quality of EO of lavender is based on the high content of both linalool and linalyl acetate [117]. Other predominant compounds include terpenes (borneol, limonene, camphene, eucalyptol, β-ocimene, 1,8-cineol, and camphor), as well as non-terpenoid aliphatic components (octenylacetate, octanon, octenol, and octanol) [115].

*Lavandula angustifolia* showed efficiency against pathogenic isolates from milk (*Staphylococcus* spp., *Bacillus cereus*, and *Kytococcus sedentarius*) [85].

*Lavandula stoechas* reduced significantly the bacterial population of *Staphylococcus aureus*, *Streptococcus agalactiae*, and *Escherichia coli*, with MIC values of 4.37, 3.75, and 3.12% *v*/*v*, respectively [86].

### 3.6. Minthostachys verticillata

*Minthostachys verticillata* (Griseb) Epling, belonging to the genus of *Minthostachys* (Benth.) Spach (Lamiaceae), is an aromatic medicinal herb, native of Argentina. It is most used in folk medicine thanks to its different therapeutic properties [89]. The chemical composition of *M. verticillate* EO is represented mainly from the monoterpenes pulegone and menthone, in addition to others, present in a lower percentage, such as isomenthone, limonene, menthol, and α-pinene [118] (Figure 6).

Montironi et al. [90] demonstrated that the EO of this species and one of its compounds, limonene, inhibited the growth of *Streptococcus uberis* causing bovine mastitis. Later the same group of researchers [89], showed that the same compounds also have antibacterial and antibiofilm effects on mastitis pathogens such as *Escherichia coli*, *Bacillus pumilus*, and *Enterococcus faecium*.

Lower concentrations were needed to inhibit most of the Gram-positive bacteria than the ones required for the Gram-negative, showing that Gram-positive bacteria were more sensitive than Gram-negative.

### 3.7. Essential Oils Derived from Other Plants

The genus *Citrus* (Rutaceae) is native of Southeast Asia; however, for their excellent source of vitamins, especially vitamin C, *Citrus* plants are grown widely all over the world [119]. The most vital byproduct of citrus processing are the EOs, which are broadly used, since they have been classified as generally recognized as safe (GRAS) [120]. Terpenes, flavonoids, carotenes, and coumarins are thought to be responsible for the strong anti-oxidative and antimicrobial activities. There is a great variation in the proportion of the main components of the EOs from one specie to another; however. the major components are d-limonene, linalyl acetate, linalool, β-Pinene, γ-Terpinene, α-Terpineol, and (E)-β-Ocimene [119].

Both *Citrus bergamia* and *C. paradisi* revealed an anti-algae and antifungal activity against *Prototheca zopfii, P. blaschkeae, P. wickerhamii, Candida parapsilosis,* and *C. krusei,* respectively [87,96]. Limonene, such as thymol, can modify fatty acid composition of cell membranes, as well as to affect their respiration and permeability. Furthermore, it produces damage to fungal cell walls, inhibiting pectin methyl esterase and cellulase [96].

*Cymbopogon* plants (Andropoganeae family) are diverse in terms of names, species, and uses, with almost all of them being aromatic and having geraniol and citral as the two major constituents of the essential oil [121]. Lemongrass and citronella grass EOs were tested, respectively, against MRSA and *E.coli* from bovine mastitis [91,122,123]. Lemongrass showed both synergistic and additive interaction with amoxicillin and norfloxacin [91]. After a treatment at 1 × MIC with *C. nardus* nanoemulsified EO, the majority of bacteria treated were irregular and withered to varying degrees, and moreover, some burst in the membrane were detected, with greater effects to the cell membrane after a treatment at 2 × MIC [123].

*Mentha* genus (family Lamiaceae) EOs are rich in oxygenated monoterpenes. Carvone, limonene, and 1,8-cineole have been identified as the primary components of Mentha spicata L. (spearmint) EO [124]. Menthol is a significant constituent of *Mentha piperita* L. (peppermint), contributing to the herb’s anti-oxidant and antimicrobial activity [125]. *Mentha spicata* EO exhibited an antibacterial activity against E.coli, causing alterations in the morphology of microbial cells [123]. On the other hand, *E. coli* and others Gram-negative strains were not susceptible to *M. piperita* EO, while the EO was effective against Gram-positive strains (*Streptococcus* spp. and *Staphylococcus* spp.) [95].

*Rosmarinus officinalis* L., rosemary, belongs to the family Lamiaceae and originated from the Mediterranean region. The EO possesses antioxidant and antimicrobial activities, attributed mainly to monoterpenes, like 1,8-cineole, borneol, pinene, limonene, camphene, camphor, and myrcene [126]. *Rosmarinus officinalis* EO shows the highest efficacy against normal and pathogenic microflora from milk, with activity similar to the one of amoxicillin [85], as well as antifungal property against *C. albicans* [92].

*Salvia* is the largest genus of the family Lamiaceae. *Salvia officinalis* L. (Sage) EO’s main components include borneol, camphor, caryophyllene, cineole, elemene, humulene, ledene, pinene, and thujone [127]. On the contrary, the study of Zarroni et al. [128] highlighted the main components of sage were, instead, carvacrol (61.01%), thymol (20.41%), and 1R-α-pinene (7.88%). Sage exhibits a bactericidal effect against *S. aureus*, *E. coli*, and *S. agalactiae.* In addition, *Salvia sclarea* inhibited the growth of *Prototheca zopfii* [84].

Another genus of the Lamiaceae family, is *Satureja* L. The EO from *Satureja montana* L., known as mountain savory, has been testes for its antimicrobial and antioxidant activity [94]. *Satureja hortensis* L. (savory) significantly decreased the population of *S. aureus, E. coli*, and *S. agalactiae* [128].

*Syzygium aromaticum* L. (Myrtaceae), commonly clove, is an aromatic plant rich in volatile compounds and antioxidants such as eugenol, eugenyl acetate, β-caryophyllene, and α-humulene [129]. Both eugenol and *Syzygium aromaticum* EO were active towards *S. aureus* isolates; however, eugenol did not significantly contribute to the antibiofilm activity of *S. aromaticum* EO, suggesting that other component within the oil may be responsible [77]. *S. aromaticum* is also effective against *P.zopfii* isolates [84].

The MIC of *Copaifera* spp. EO ranged from 15.62 to 1000 μg/mL against *Staphylococcus* spp. and it was less than 100 μg/mL against some strains of *Corynebacterium* spp. [130].

A xanthan gum film-forming suspension incorporating *Eugenia caryophyllata* EO, the main component of which is eugenol (93.30%), was effective against *C. albicans, E. coli* and *S. aureus* [83].

Cedar (*Juniperus virginiana*), manuka (*Leptospermum scoparium*), and patchouli (*Pogostemon cablin*) exhibited slightly higher activity in comparison to Geranium (*Pelargonium graveolens*) oils against *S. aureus*, *S. epidermidis* and *S. xylosus* [82].

## 4. In Vivo Studies

Although there is an interesting number of in vitro studies to assess the efficacy of essential oils and their main components on various isolates from mastitis cows (see Section 3, transferring these results in in vivo models is always challenging. Thus, the in vivo studies on dairy cows with diagnosed subclinical and clinical forms of mastitis are very limited and less conclusive.

In addition to different types of EOs, the pharmaceutical formulations and the dosage regimens used are diverse. Essential oils can be applied intramammarily or topically by the application of semisolid pharmaceutical formulations (gel or ointment) [131].

Since the demonstrated in vitro efficacy of *M. verticillata* EO on mastitis causing pathogens [89,90], its efficacy has been tested by Montironi et al. [118] on macrophage phagocytosis, and its immunomodulatory and protective effects evaluated in a murine model of *Enterococcus faecium* mastitis. It was proved the modulating effect of the EO on the adhesion mechanisms and on the phagocytic capacity of macrophages. Furthermore, EO induced ROS production. Given the similarities between the mammary gland of mice and cows, this study offers a good model for bovine mastitis. In mammary glands of treated mice, inoculation of EO moderated the innate immune response by decreasing the infiltration of polymorphonuclear neutrophils (PMNs) and IL-1β and TNF-α mRNA expression. The expression of IL-10 was increased at 96 h of infection, while the number of activated CD4+ or CD8+ T cells, or the production of specific antibodies were not affected. The EO might play an important role in the resolution of the infection in the first few hours, without activating adaptive immunity. A marked decrease in the bacterial count was also observed in the glands of EO-treated group.

Essential oil of *M. verticillate*, limonene (L), and their spray-drying microencapsulations (McEO, McL) were able to stimulate the production of *E. faecium*-specific IgG, with opsonizing capacity, in Balb/C mice in a similar way to Incomplete Freund’s Adjuvant. The proportion of CD4+ and CD8+ T cells producers of IFN-γ was also increased. The proliferation of cells of mice inoculated with vaccine formulations containing McEO and McL was more efficient than cells from saline-solution-treated mice. Microencapsulation resulted as an effective strategy to increase the adjuvant potential of EO or L [132].

In a murine model of lipopolysaccharide (LPS)-induced mastitis, *Houttuynia cordata* Thunb EO (HEO) and a self-microemulsion preparation of the same EO (SME-HEO), significantly downregulated pro-inflammatory factors TNF-α and IL-1β, upregulated anti-inflammatory factor IL-10, inhibited myeloperoxidase (MPO) expression, and alleviated histopathological injury in murine mammary gland tissues in a dose-dependent manner. They also improved the integrity of the blood–milk barrier (BMB) by upregulating the expression of junction proteins: ZO-1, claudin-1, claudin-3, and occluding. HEO plays an anti-inflammatory role by blocking the activation of MAPK and other inflammatory signaling pathways, as well as the expression of iNOS. The authors concluded that HEO might provide a novel treatment for mastitis [133].

After testing the in vitro efficacy of *Origanum vulgare* EO against methicillin-resistant *Staphylococcus aureus* (MRSA) and methicillin-susceptible *S. aureus* (MSSA), Hamlaoui et al. [134] studied, under in vivo condition, its effect on the bacteriological quality of raw milk from cows affected by subclinical mastitis. EO was topically applied to cows twice a day for three days. After the application, a significant decrease in the number of total aerobic mesophilic bacteria (TMAB) was observed. For coagulase-positive staphylococci (CPS), there was a significant absence in 40% of the unit component samples after the application of the EO, even if the decrease effect on the mean number of CPS in the samples from healthy and affected quarters was statistically nonsignificant.

In a previous study, Cho et al., in 2015, showed that intramammary treatment with an ointment containing 0.9 mL of oregano EO reduced bovine mastitis with caused by *S. aureus* and *E. coli*, compared to gentamicin. An improvement in udder conditions after the treatment was observed, with both SCCs and white blood cells (WBCs) significantly decreasing when compared with those pre-treatment [135].

Two gel products (R4 and R7), containing essential oil of oregano, lavender, and rosemary at different concentrations, were administered intramammarily to cows with clinical and subclinical mastitis, resulting in a decreased number of SCCs and a significant lower mean number of bacteria in all treated animals. R7, with higher concentrations of all the active ingredients, proved to be more effective than R4, being efficacious in six cows out of eight, while the latter being valuable in three out of eight cows [136].

The therapeutic efficacy of *Eucalyptus globulus* and *Lavandula hybrida* essential oils as teat dips in subclinical mastitis cows was evaluated in a clinical trial on 24 animals. Teat dip preparations were arranged at 98% and 96% for both the EOs and applied twice a day for 28 days. The parameters of milk pH, somatic cell count, and colony forming unit were measured. After treatment, milk pH and SCC were significantly (*p* < 0.05) decreased in all four EOs groups, and the values were restored to normal at twenty-eighth-day post-application. Milk yield, one of the most important economic losses correlated to subclinical mastitis, showed an increasing trend in essential oil treated groups. Moreover, Eos significantly reduced the colony-forming units [137].

A novel EO-based pharmaceutical formulation (Phyto-Bomat) was administered intramammarily twice a day for 5 days to mastitis cows. Phyto-Bomat contains Eos of *Thymus vulgaris* L. *and Thymus serpyllum* L., *Origanum vulgare* L., and *Satureja montana* L., the dominant compounds were thymol and carvacrol, at 12.58 ± 1.23 mg/mL and 23.11 ± 2.31 mg/mL, respectively.

Antimicrobial activity was the highest against Gram-positive strains. Since EOs could confer an undesirable odor or taste to milk or dairy products, the withdrawal period was determined. Blood and milk samples were analyzed for thymol and carvacrol residues using gas-chromatography and mass spectrometry (GC–MS). The concentration of thymol and carvacrol in milk samples after 24 h from the administration was at the same level as before application. On the other hand, thymol and carvacrol were still detectable in plasma samples even after 24 h post-treatment, with values ranging from 0.15 to 0.38 and 0.21 to 0.66 μg/mL, respectively [138].

Successively, the same group of researchers evaluated the effectiveness of Phyto-Bomat as an alternative to the existing treatment with antibiotics. The therapeutic response of the proposed formulation administered to the mastitis cows was monitored using clinical observations and bacteriological analyses. The EO formulation apparently prevented the subclinical cases of mastitis to develop the clinical form. Treatment with Phyto-Bomat did not exert any local or systemic side effects in any of the cows. No differences between treatment with antibiotic (cephalexin) and Phyto-Bomat were found in clinical outcomes, suggesting that this formulation can be proposed as an alternative for bovine mastitis therapy. However, bacteriological cure after Phyto-Bomat administration showed negative results since it seems to be ineffective against *E. coli* and *S. marcescens* [139].

## 5. Future Prospects and Conclusions

The impact of bovine mastitis worldwide is extremely high due to the reduced yield and poor quality of milk. Bovine mastitis represents a major disease in the dairy industry that leads to huge economic losses, reaching a global annual scale of approximately USD 35 billion [4]. The costs related to mastitis include direct and indirect losses, since the disease usually induces permanent and/or irreversible damage to milk-producing glandular tissues. Moreover, losses due to subclinical mastitis are three times more severe than those due to clinical cases [30], because they are difficult to quantify as they are not visible to farmers [31].

The current approach to control and inhibit mastitis progression remains the use of antibiotics, although its efficacy is limited and bacteriological elimination is often less than 60% [39]; in addition, the development of antibiotic-resistant strains of pathogens, and more generally, of non-pathogenic bacteria (e.g., commensal bacteria) have become a critical challenge [40].

Plant-derived EOs and some of their components have considerable potential for the safe treatment and prevention of bovine mastitis. Some of the major EO components, such as EO-phenols (thymol and carvacrol), monoterpenes (limonene and p-cymene), aldehydes (cinnamaldehyde), phenyl-methyl ethers (eugenol), which occur in diverse plant species, have proven to be effective against a range of mastitis causing pathogens.

However, only limited clinical studies are available so far. Essential oils are complex mixtures of natural products and they do not fit well into the classical drug discovery model.

On the other hand, EOs possess several unusual properties, which may prove advantageous for therapy. The lipophilic nature of EOs and their components allows them to cross the cell membrane of bacteria and mitochondria, destabilizing cellular architecture, leading to disruption of membrane integrity and increased permeability. In addition, EOs increase the adhesion mechanisms and the phagocytic capacity of macrophages, as well as ROS production and improve the integrity of the blood-milk barrier by upregulating the expression of junction proteins. For these reasons, EOs and their components could serve as guiding compounds to develop an integrated or alternative therapy to the antibiotic treatment.

This perspective is hampered by the limited number of in vivo studies. Moreover, in many cases, the molecules used for in vivo studies have not been tested for the presence of lipopolysaccharide or other pro-inflammatory molecules, which may affect the experimental design, particularly when infused directly into the mammary gland. Most in vivo studies, including treatment with an external application, do not provide sufficient scientific evidence to support the use of EO directly into the mammary gland [140].

Alternative treatments must also be sustainable from an economic point of view in order to become a real strategy against bovine mastitis. Although is not easy to obtain the data on nonmonetary units, i.e., health benefits, such as the mortality rate, prolonged life/lactation period and improvements in the quality of life, economic evaluations should be applied to both conventional antibiotics and proposed treatment using EOs.

The presence of antimicrobial residues in milk is one of the biggest challenges of the food and veterinary industries worldwide since they could interfere with the production of dairy products and may cause developing resistance. Costs for rejecting milk due to the withdrawal period could be bigger than the cost of the treatment, thus more in vivo studies on the withdrawal period of EO-based pharmaceutical formulations in bovine mastitis treatment are essential to establish the effectiveness of EOs.

Most studies focus on the major constituents of EOs, but some results indicate that EOs are more active than the individual components, so minor constituents and synergism of action among the various constituents represent an interesting area for future research. Finally, the application of nanotechnology (preparation of nanoemulsions with EOs) could help solve problems such as solubility, release and permeation, bioavailability, and stability, which may limit the therapeutic use of EOs in biological systems.

## Figures and Tables

**Figure 1 molecules-28-03425-f001:**
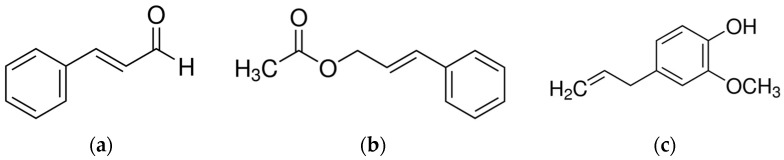
Main components of Cinnamon EOs. (**a**) trans-Cinnamaldehyde; (**b**) Cinnamyl acetate; (**c**) Eugenol.

**Figure 2 molecules-28-03425-f002:**
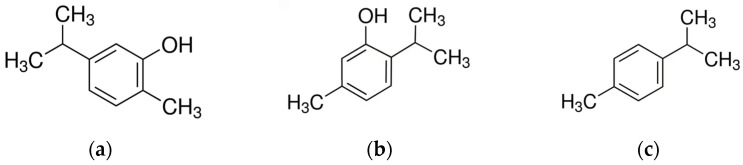
Main components of *Origanum* EOs. (**a**) Carvacrol; (**b**) Thymol; (**c**) p-Cymene.

**Figure 3 molecules-28-03425-f003:**
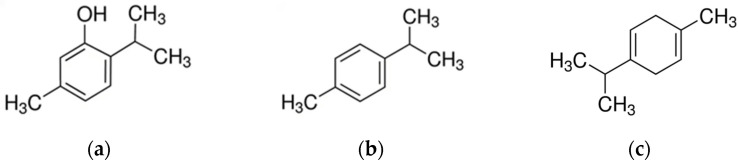
Main components of *Thymus* EOs. (**a**) Thymol; (**b**) p-Cymene; (**c**) γ-Terpinene.

**Figure 4 molecules-28-03425-f004:**
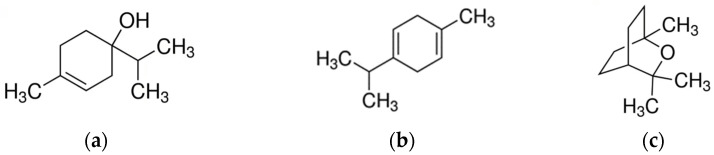
Main components of *Melaleuca* EOs. (**a**) Terpinen-4-ol; (**b**) γ-Terpinene; (**c**) 1,8-Cineole.

**Figure 5 molecules-28-03425-f005:**
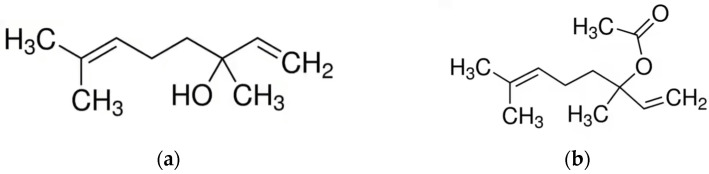
Main components of *Lavandula* Eos. (**a**) Linalool; (**b**) Linalyl acetate.

**Figure 6 molecules-28-03425-f006:**
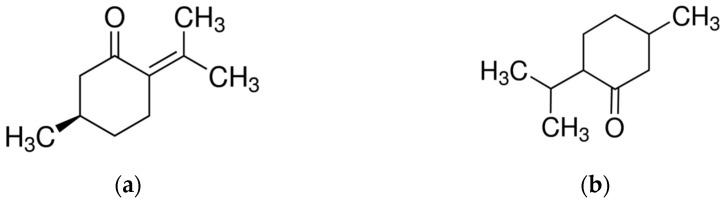
Main components of *Minthostachys verticillata* EOs. (**a**) Pulegone; (**b**) Menthone.

**Table 1 molecules-28-03425-t001:** List of selected essential oils and their major constituents with in vitro activity (for an expanded list, see Appendix A).

Name of the Essential Oil	Main Component(s)	Tested Microorganisms	MIC	MBC	Zone Inhibition (mm)	Biofilm (%)	Reference
-	Carvacrol	*S*. *aureus*	0.38 ± 0.00 mg/mL *	0.75 ± 0.00 mg/mL *	32.7 ± 3.01		[74]
-	Carvacrol	*E*. *coli*	0.75 ± 0.00 mg/mL *	1.25 ± 0.43 mg/mL *	31.7 ± 0.58		[74]
-	Carvacrol	*Klebsiella pneumoniae*	0.75 ± 0.00 mg/mL *	1.5 ± 0.00 mg/mL *	13.7 ± 0.58		[74]
-	Carvacrol	*C*. *albicans*	0.1–0.2 mg/mL	0.4–0.8 mg/mL			[75]
-	Carvacrol	*Staphylococcus* spp., *Streptococcus* spp.	0.02–0.4 mg/mL	0.2–0.78 mg/mL			[75]
-	Carvacrol	*E*. *coli*, *Klebsiella pneumoniae*	0.02–0.2 mg/mL	0.1–0.7 mg/mL			[75]
-	Carvacrol	*Staphylococcus* spp.	584 µg/mL	732 µg/mL			[76]
-	Cinnamaldehyde	*S*. *aureus*	0.199 mg/mL			69.4/44.9	[77]
-	trans-Cinnamaldehyde	*S*. *aureus*	0.3 ± 0.00 mg/mL *	0.66 ± 0.00 mg/mL *	25.3 ± 1.53		[74]
-	trans-Cinnamaldehyde	*E*. *coli*	0.6 ± 0.00 mg/mL *	1.32 ± 0.00 mg/mL *	20 ± 1.00		[74]
-	trans-Cinnamaldehyde	*Klebsiella pneumoniae*	0.6 ± 0.00 mg/mL *	1.32 ± 0.00 mg/mL *	11.0 ± 1.00		[74]
-	trans-Cinnamaldehyde	*Staphylococcus* spp.	598 mg/L	2238 mg/L			[78]
*Cinnamon*	/	*S*. *agalactiae*	0.01 ± 0.01 μg/mL **		20.57–38.29	50%	[79]
*Cinnamomum aromaticum*	e-Cinnamaldehyde (94.67%)	*Staphylococcus* spp.	0.625 μL/mL	1.25–10 μL/mL			[80]
*Cinnamomum aromaticum*	e-Cinnamaldehyde (94.67%)	*Candida* spp., *Cryptococcus* spp., *Rhodotorula glutinis*	0.625–1.25 μL/mL	1.25–2.5 μL/mL			[80]
*Cinnamomum cassia*	Cinnamaldehyde (59.96%)	*S*. *aureus*, *S*. *epidermidis*, *S*. *xylosus*	0.0125% *v*/*v*	0.05% *v*/*v*	29.6, 19.1, 33.3		[81]
*Cinnamomum cassia*	Cinnamaldehyde (59.96%)	*S*. *hyicus*	0.00625% *v*/*v*	0.025% *v*/*v*	27.0		[81]
*Cinnamomum cassia*	Cinnamaldehyde (59.96%)	*E*. *coli*	0.025% *v*/*v*	0.10% *v*/*v*	30.7		[81]
*Cinnamomum cassia*	/	*S*. *aureus*, *S*. *epidermidis*, *S*. *xylosus*	0.078–1.250% *v*/*v*	0.039–1.250% *v*/*v*			[82]
*Cinnamomum cassia*	Cinnamaldehyde (98.29%)	*C*. *albicans*	1 wt%		5.0–25.0		[83]
*Cinnamomum cassia*	Cinnamaldehyde (98.29%)	*E*. *coli*	0.5–1 wt%		3.0–21.0		[83]
*Cinnamomum cassia*	Cinnamaldehyde (98.29%)	*S*. *aureus*	0.5 wt%		7.0–32.0		[83]
*Cinnamomum zeylanicum*	Cinnamaldehyde (86.59%)	*S*. *aureus*	0.243 mg/mL			74.7/45.3	[77]
*Cinnamomum zeylanicum* Ness	Cinnamaldehyde (76.8%), Methoxycinnamaldehyde (11.7%)	*Prototheca zopfii*	0.2–0.4 μL/mL				[84]
*Cinnamomum zeylanicum*	Isoeugenol (96%)	*Staphylococcus* spp.	2032 mg/L	4263 mg/L			[78]
*Lavandula angustifolia*	/	*Staphylococcus* spp., *Kytococcus sedentarius*, *Bacillus cereus*			15.17 ± 5.06		[85]
*Lavandula stoechas*	17-Pentatriacontene (42.15%), Linalyl acetate (26.82%), Eucalyptol (18.87%)	*E*. *coli*	3.12% *v*/*v*	6.25% *v*/*v*			[86]
*Lavandula stoechas*	17-Pentatriacontene (42.15%), Linalyl acetate (26.82%), Eucalyptol (18.87%)	*S*. *aureus*	4.37% *v*/*v*	8.75% *v*/*v*			[86]
*Lavandula stoechas*	17-Pentatriacontene (42.15%), Linalyl acetate (26.82%), Eucalyptol (18.87%)	*S*. *agalactiae*	3.75% *v*/*v*	7.50% *v*/*v*			[86]
*Melaleuca alternifolia*	Terpinen-4-ol (39.1%), γ-Terpinene (21.1%), α-Terpinene (9.2%)	*C*. *albicans*	6.25 mg/mL	6.25–12.50 mg/mL			[75]
*Melaleuca alternifolia*	Terpinen-4-ol (39.1%), γ-Terpinene (21.1%), α-Terpinene (9.2%)	*Staphylococcus* spp., *Streptococcus* spp.	3.13–25 mg/mL	6.25–52 mg/mL			[75]
*Melaleuca alternifolia*	Terpinen-4-ol (39.1%), γ-Terpinene (21.1%), α-Terpinene (9.2%)	*E*. *coli*, *Klebsiella pneumoniae*	0.78–3.13 mg/mL	6.25–12.50 mg/mL			[75]
*Melaleuca alternifolia*	/	*Prototheca zopfii*, *P*. *wickerhamii*	0.03–0.12%				[87]
*Melaleuca alternifolia*	/	*Candida parapsilosis*, *Candida krusei*	0.06–0.25%				[87]
*Melaleuca armillaris*	1,8-Cineole (72.3%), Limonene (7.8%)	*S*. *aureus*	6.25–25 µL/mL	12.5–50 µL/mL			[88]
*Melaleuca armillaris*	1,8-Cineole (72.3%), Limonene (7.8%)	*S*. *aureus*	3.1–25 µL/mL	12.5–50 µL/mL			[88]
*Minthostachys verticillata*	Pulegone (74.96%), Menthone (20.38%)	*E*. *coli*	0.90–14.51 mg/mL	/		36.51–89.60	[89]
*Minthostachys verticillata*	Pulegone (74.96%), Menthone (20.38%)	*Enterococcus faecium*	3.63 mg/mL	29 mg/mL		36.51–89.60	[89]
*Minthostachys verticillata*	Pulegone (74.96%), Menthone (20.38%)	*Bacillus pumilus*	1.8–29 mg/mL	/		36.51–89.60	[89]
*Minthostachys verticillata*	Pulegone (51.7%), Menthone (37.8%)	*Streptococcus uberis*	14.3–114.5 mg/mL	114.5–229 mg/mL		23.50–88.25	[90]
*Oregano*	Carvacrol (68.78%), o-cymene (9.80%)	*S*. *aureus (MRSA)*	100–4000 μg/mL		2.5 ± 0.5–9.5 ± 1.5		[91]
*Origanum floribundum* Munby.	Thymol (50.47%), p-Cymene (24.22%), γ-Terpinene (11.27%)	*C*. *albicans*	17.18–23.14% µg/mL (MIC 80%)				[92]
*Origanum majorana* L.	trans-Sabinene hydrate (38.10%), Borneol (11.10%), Sabinene (7.60%)	*Prototheca zopfii*	0.25–0.5 µL/mL				[93]
*Origanum majorana*	3-Cyclohexene-1-ol,4-methyl-1-(1-methylethyl)-,(R)-(44.84%), α-Terpineol (6.83%), p-Cymene (6.75%)	*E*. *coli*	1.56% *v*/*v*	3.12% *v*/*v*			[86]
*Origanum majorana*	3-Cyclohexene-1-ol,4-methyl-1-(1-methylethyl)-,(R)-(44.84%), α-Terpineol (6.83%), p-Cymene (6.75%)	*S*. *aureus*	0.62% *v*/*v*	1.25% *v*/*v*			[86]
*Origanum majorana*	3-Cyclohexene-1-ol,4-methyl-1-(1-methylethyl)-,(R)-(44.84%), α-Terpineol (6.83%), p-Cymene (6.75%)	*S*. *agalactiae*	1.87% *v*/*v*	3.75% *v*/*v*			[86]
*Origanum vulgare*	Carvacrol (92%)	*Staphylococcus* spp.	1600 µg/mL	2288 µg/mL			[76]
*Origanum vulgare* L.	Carvacrol (80.35%), p-Cymene (4.82%), Thymol (4.21%)	*Streptococcus* spp., *E*. *coli*, *Cronobacter sakazakii*, *Klebsiella oxytoca*, *Staphylococcus* spp.	0.78–6.25 mg/mL,	1.56–12.5 mg/mL			[94]
*Origanum vulgare* L.	Carvacrol (77.29%), p-Cymene (8.85%), γ-Terpinene (4.96%)	*Prototheca zopfii*	0.5–1 µL/mL				[93]
*Origanum vulgare* L.	Carvacrol (78.94%), Thymol (4.87%), p-Cymene (4.52%)	*Proteus mirabilis*	3.125 ± 1.35 mg/mL *	3.125 ± 1.35 mg/mL *			[95]
*Origanum vulgare* L.	Carvacrol (78.94%), Thymol (4.87%), p-Cymene (4.52%)	*Serratia marcescens*	3.125 ± 1.91 mg/mL *	6.25 ± 3.83 mg/mL *			[95]
*Origanum vulgare*	Carvacrol (90.50%)	*C*. *albicans*	0.3 wt%		3.0–25.0		[83]
*Origanum vulgare*	Carvacrol (90.50%)	*E*. *coli*	0.5 wt%		3.0–23.5		[83]
*Origanum vulgare*	Carvacrol (90.50%)	*S*. *aureus*	0.3–0.5 wt%		5.0–24.5		[83]
*Origanum vulgare*	Carvacrol (65.9%), p-Cymene (9.3%)	*P*. *zopfii*, *P*. *blaschkeae*	0.75/1%				[96]
-	Thymol	*Staphylococcus* spp.	429.68 ± 123.53 µg/mL	859.38 ± 247.05 µg/mL			[97]
-	Thymol	*Streptococcus* spp.	664.06 ± 370.58 µg/mL	1328.13 ± 741.16 µg/mL			[97]
-	Thymol	*E*. *coli*	976.55 ± 887.91 µg/mL	1953.1 ± 1775.82 µg/mL			[97]
-	Thymol	*C*. *albicans*	0.05–0.4 mg/mL	0.4–1.6 mg/mL	43.8 ± 0.2		[75]
-	Thymol	*Staphylococcus* spp., *Streptococcus* spp.	0.1–0.2 mg/mL	0.2–0.4 mg/mL	42.2 ± 5.5		[75]
-	Thymol	*E*. *coli*, *Klebsiella pneumoniae*	0.1–0.2 mg/mL	0.4 mg/mL	31.2 ± 0.7		[75]
-	Thymol	*S*. *aureus*	0.75 ± 0.00 mg/mL *	1.5 ± 0.00 mg/mL *	24.3 ± 2.00		[74]
-	Thymol	*E*. *coli*	0.38 ± 0.00 mg/mL *	0.63 ± 0.21 mg/mL *	20.7 ± 1.16		[74]
-	Thymol	*Klebsiella pneumoniae*	0.75 ± 0.00 mg/mL *	1.5 ± 0.00 mg/mL *	13.5 ± 0.50		[74]
-	Thymol	*Staphylococcus* spp.	427 µg/mL	856 µg/mL			[76]
*Thymus ciliatus* Desf.	Thymol (62.41%), p-Cymene (15.51%), Carvacrol (6.12%)	*C*. *albicans*	15.02–20.96 µg/mL (MIC 80%)				[92]
*Thymus fontanesii* Boiss. Et Reut.	Carvacrol (62.25%)	*E*. *coli*	1/1600 µL/mL	0.625 µL/mL	25.33 ± 1.53		[98]
*Thymus fontanesii* Boiss. Et Reut.	Carvacrol (62.25%)	*S*. *aureus*	1/1600 µL/mL	0.625 µL/mL	35 ± 0.7		[98]
*Thymus serpyllum*	Thymol (54.17%), γ-Terpinene (22.18%), p-Cymene (16.66%)	*Streptococcus* spp.	0.78–3.125 mg/mL	1.56–6.25 mg/mL			[99]
*Thymus serpyllum*	Thymol (54.17%), γ-Terpinene (22.18%), p-Cymene (16.66%)	*E*. *coli*, *Enterobacter sakazakii*	6.25 mg/mL	12.5 mg/mL			[99]
*Thymus serpyllum*	Thymol (54.17%), γ-Terpinene (22.18%), p-Cymene (16.66%)	*Klebsiella oxytoca*, *Staphylococcus* spp.	3.125 mg/mL	6.25 mg/mL			[99]
*Thymus serpyllum* L.	Thymol (55.11%), γ-Terpinene (22.31%), p-Cymene (16.66%)	*Proteus mirabilis*	3.125 ± 1.35 mg/mL *	6.25 ± 2.7 mg/mL *			[95]
*Thymus serpyllum* L.	Thymol (55.11%), γ-Terpinene (22.31%), p-Cymene (16.66%)	*Serratia marcescens*	1.56 ± 0.96 mg/mL *	3.125 ± 1.91 mg/mL *			[95]
*Thymus vulgaris*	Thymol (45.22%), p-Cymene (23.83%)	*Streptococcus* spp.	0.39–1.56 mg/mL	0.78–6.25 mg/mL			[99]
*Thymus vulgaris*	Thymol (45.22%), p-Cymene (23.83%)	*E*. *coli*, *Enterobacter sakazakii*	3.125 mg/mL	6.25 mg/mL			[99]
*Thymus vulgaris*	Thymol (45.22%), p-Cymene (23.83%)	*Klebsiella oxytoca*, *Staphylococcus* spp.	1.56 mg/mL/6.25 mg/mL	6.25 mg/mL/12.5 mg/mL			[99]
*Thymus vulgaris* L.	Thymol (52.96%), p-Cymene (17.73%), γ-Terpinene (5.97%)	*Prototheca zopfii*	0.25–1 µL/mL				[93]
*Thymus vulgaris* L.	Thymol (38.1%), p-Cymene (29.1%), γ-Terpinene (5.2%)	*Prototheca zopfii*	0.6–1.0 μL/mL				[84]
*Thymus vulgaris*	Thymol (52.6%), p-Cymene (15.3%)	*P*. *zopfii*, *P*. *blaschkeae*	0.75/1%				[96]
*Thymus vulgaris*	/	*S*. *aureus*, *S*. *chromogenes*, *S*. *uberis*	2% *v*/*v*				[100]
*Thymus vulgaris*	/	*S*. *aureus*	5%	5%	17.83–37.5		[26]
*Thymus vulgaris*	/	*Streptococcus agalactiae*	4%	5%	13.5–30.83		[26]
*Thymus vulgaris*	/	*E*. *coli*	1%	1%	14.66–34.33		[26]
*Thymus vulgaris*	γ-Terpinene (64%)	*Staphylococcus* spp.	1564 µg/mL	2370 µg/mL			[76]
*Thymus vulgaris* L.	Thymol (46.37%), p-Cymene (23.83%), γ-Terpinene (3.46%)	*Proteus mirabilis*	3.125 ± 0.00 mg/mL *	6.25 ± 2.7 mg/mL *			[95]
*Thymus vulgaris* L.	Thymol (46.37%), p-Cymene (23.83%), γ-Terpinene (3.46%)	*Serratia marcescens*	1.56 ± 0.96 mg/mL *	3.125 ± 1.91 mg/mL *			[95]
*Thymus vulgaris*	/	*S*. *aureus*, *S*. *epidermidis*, *S*. *xylosus*	0.010–0.625 *v*/*v*%	0.010–0.625 *v*/*v*%			[82]

- Pure compound tested; / Data not available; * ± SD; ** ± SE.

## Data Availability

No new data were created or analyzed in this study. Data sharing is not applicable to this article.

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
