# Peer review of "Plant Essential Oils as a Tool in the Control of Bovine Mastitis: An Update"

_molecules, 2023, doi:10.3390/molecules28083425_

Round 1

Reviewer 1 Report

The authors have done an interesting job of summarising what is available on the subject by presenting, in a concise but exhaustive manner, both the starting problem, linked to the need to remedy the emergence of antibiotic-resistant pathologies, and the possible resolution through the study of essential oils containing volatile compounds potentially active against the bacteria responsible for these pathologies.I therefore recommend tpublication of the paper in present form, subject to checking some typos in the text (for example, b instead of beta-caryophyllene at line 261, or terpinene-4-ol instead terpinen-4-ol at line 303.)

Reviewer 2 Report

Manuscript entitled “L Plant Essential Oils as a Tool in the Control of Bovine Mastitis: An Update” . This article have good collections that provide an updated overview of the in vitro and in vivo studies available on EOs and their main components as an antibacterial treatment against a variety of mastitis causing pathogens.

Introduction in the paper is too short.

Chemical structures of active essential oils part can be added

No figures and flow  chart for better undrestanding of contents.

There should be added 2-3 more images and tables.

Overall manuscript is well written but I can be improved.

Reviewer 3 Report

GENERAL COMMENTS: The subject addressed in this review is very relevant for dairy industry worldwide. The authors included statistical data relevant to the subject and I believe after minor revision it could be suitable for publishing.

Line 8: Bovine mastitis, inflammation of the mammary gland… I suggest deleting inflammation of the mammary gland because is redundant, I strongly believe that whoever reads this article sure knows what mastitis means.

Line 13. causing limited bacteriological elimination in mastitis treatment, as well as a serious threat for public healthI suggest: resulting in a limited resolution of mastitis treatments, as well as a serious threat for public health.

Line 14: Alternatives, especially those based on natural products such as plant essential oils (EOs), to antibiotic therapy are needed, in particular against multidrug-resistant bacteria…. I suggest: Novel alternatives like the use of plant essential oils (EOs) are needed to replace antibiotic therapy when facing multidrug-resistant bacteria.

Lines 17, 18, 19, 241, 245, 262, 271, 546, 550 …etc: in vitro an in vivo – use Italics.

Line 20: further clinical researches- I suggest to change the word researches for trials

Line 31: change word classes for categories.

Line 46: Contrarily, environmental pathogens do not usually live on the cow’s udder and teat skin, they are opportunistic invaders… I suggest: Contrarily, environmental pathogens are not part of the mammary gland microbiota, they are opportunistic invaders.

Line 103: or intravenous infusion… infusions

Line 103: DCT … Abbreviations should not be used at the beginning of a sentence. Check throughout the manuscript ex. Lines 177 and 184, 186, 198, 207, 448, 662 etc…

Line 113: is advisable on the advice of the veterinarian… is suitable if the veterinarian advice to.

Line 115: main concern… I suggest changing for: main objective.

Line 144: Hence, it is necessary to look for alternatives to antibiotic therapy, especially against multidrug-resistant bacteria, particularly those based on natural products such as plants… Suggestion: Hence, it is necessary to look for alternative treatments based on natural products such as plants, especially when facing multidrug-resistant bacteria.

Line 146: researchers all over the world are looking for finding different alternatives to the use of antibiotics… suggestion: researchers all over the world are working to find novel alternatives to the use of antibiotics.

Line 150: traditional medicines- traditional medicine.

Line 152: Many medicinal plants, with different biological properties, are already used in this field, although data on the efficacy of these treatments and reports on pharmacological studies, especially in vivo, are limited [54]…. Suggestion: Many medicinal plants with different biological properties are already used in this field, although in vivo trials reporting their efficacy, pharmacokinetics and pharmacodynamics are scarce.

Lines 154-156: Compared to antibiotics, compounds of plant origin have the advantage of making the emergence of resistance more difficult, which is why it is important to validate their use with rigorous experimental studies… I believe this statement is wrong, as the authors cited state: One explanation for the failure to identify potent broad-spectrum plant-derived antibacterials is that plants may use a different chemical strategy for the control of microbial infections, perhaps to decrease the selective pressure for developing antibiotic resistance… it refers to a plant-derived product containing a group of secondary metabolites or phytochemicals that act as antimicrobial agents through different mechanism alone or with synergistic interactions. I suggest rewriting this idea. I suggest the authors to check this reference, it may help with their idea: Jubair et al., 2021. Review on the Antibacterial Mechanism of Plant-Derived Compounds against Multidrug-Resistant Bacteria (MDR).  https://doi.org/10.1155/2021/3663315

Lines 165-167: EOs are formed by aromatic plants as secondary metabolites and they play an important role in the protection of plants as antibacterials, antivirals, antifungals, insecticides, and also against herbivores by reducing their palatability for such plants…. Again, the information is misinterpreted: is not of aromatic plants that EOs are formed of, and secondary metabolites are not aromatic plants as presented in that sentence. Many EOs are obtained from aromatic plants which contain secondary metabolites or phytochemicals, which play that role in the protection of plants… I would like to suggest complementary reading: Shelepova et al., 2022. Aromatic Plants Metabolic Engineering: A Review. https://doi.org/10.3390/agronomy12123131

Line 662: change word thus for so.

Round 2

Reviewer 2 Report

Manuscript is revised substantially by authors. So it can be accepted in its present form.